# Longitudinal Associations of Physical Activity Patterns and the Environment: An 18-Year Follow-Up to the MESA Study

**DOI:** 10.3390/ijerph191710925

**Published:** 2022-09-01

**Authors:** Maíra Tristão Parra, Augusto César Ferreira De Moraes, Marcus Vinicius Nascimento-Ferreira, Paul J. Mills, Matthew Allison

**Affiliations:** 1Hebert Wertheim School of Public Health and Longevity Science, UC San Diego, San Diego, CA 92093, USA; 2Michael & Susan Dell Center for Healthy Living, The University of Texas Health Science Center, Houston School of Public Health (UTHealth School of Public Health), Department of Epidemiology, Human Genetics and Environmental Science, Austin Campus, Austin, TX 78701, USA; 3Department of Epidemiology, School of Public Health, University of São Paulo, São Paulo 05508-060, Brazil; 4YCARE (Youth/Child cArdiovascular Risk and Environmental) Research Group, Faculdade de Medicina, Universidade de Sao Paulo, São Paulo 05508-060, Brazil; 5Physical Activity and Behavior Research (HEALTHY-BRA) Group, Universidade Federal do Tocantins, Miracema 77650-000, Brazil; 6Department of Preventive Medicine, School of Medicine, UC San Diego, San Diego, CA 92093, USA

**Keywords:** cohort studies, physical activity, environment, walkability, perceived environment

## Abstract

**Introduction:** Cross-sectional association between the neighborhood-built environment and physical activity (PA) has been demonstrated previously, indicating the importance of neighborhood perception characteristics such as walkability, safety, and the connectivity of streets on PA levels. Our study aimed to assess the longitudinal data from participants of the Multi-Ethnic Study of Atherosclerosis (MESA) to evaluate the potential relationship between perceived environment and PA patterns. **Methods:** We analyzed data from a subset of participants (n = 3097) with available PA data who participated in a prospective cohort conducted from 2000 to 2018. The exposure variables were the perceived aspects of the neighborhood environment and the perception of safety, and the outcome was patterns of PA. Patterns were defined as categories reflecting meeting versus not meeting PA guidelines over time. We created the following categories: adopters (individuals who did not meet guidelines at baseline but met guidelines at Exam 6), relapsers (individuals who met guidelines at baseline but did not meet guidelines at Exam 6), maintainers (individuals who met guidelines both at baseline and Exam 6), and insufficiently active (individuals who did not meet guidelines at either baseline or Exam 6). The maintainers’ group was considered the reference category. We estimated the relative risk to assess the magnitude effect of the association between environmental perceptions and the outcome. **Results:** Individuals who reported that lack of parks and playgrounds was “not a problem” in their neighborhood had a 2.3-times higher risk of decreasing their physical activity (i.e., the “relapser” category) compared to maintainers. After full adjustment, perceiving poor sidewalks as “somewhat a serious problem” was associated with a 64% lower risk of becoming an adopter than a maintainer. When compared to those who perceive the neighborhood as “very safe”, perception of the neighborhood as “safe” to “not at all safe” (ratings 3, 4, and 5, respectively, on the perceived safety scale) was significantly associated with being classified in the adopter category. **Conclusions:** As the first longitudinal study of the association of perceived environment and physical activity within the MESA cohort, we conclude that a few aspects are longitudinally associated with being physically active among adults.

## 1. Introduction

Key message:Neighborhood perception characteristics such as walkability, safety, and the connectivity of streets are associated with physical activity (PA) levels in adults.Only 46.5% of adults reported they continued to practice physical activity at least 150 min per week during the follow-up.The perceived environment is longitudinally associated with being physically active among adult PA.

Insufficient physical activity (PA) can contribute to premature morbidity and mortality, especially the development of chronic diseases such as coronary heart disease (CHD), type 2 diabetes, and breast and colon cancers [1], and thereby cost billions to healthcare systems worldwide [2]. The association of the built environment (home, workplace, neighborhood) and physical activity has been demonstrated previously [3], indicating the importance of objectively measured neighborhood characteristics such as walkability, safety, the connectivity of streets, as well as one’s perception of the environment. Psychosocial factors, such as perceived enjoyment of PA, perceived social support, and self-efficacy, were shown as moderators of the relationship between perceived environmental attributes and walking and recreational moderate-to-vigorous PA levels. In this regard, positive environmental perceptions were associated with higher PA levels [4].

Previous analyses from the Multi-Ethnic Study of Atherosclerosis (MESA) have demonstrated significant associations between the environment and health outcomes that are relevant to sustaining an adequate healthy lifestyle. For example, cross-sectional analyses have shown that living in areas with a high density of recreational resources for PA is positively associated with participation in these activities [5]. More contemporary longitudinal analysis confirmed that a greater density of recreational facilities was associated with less decline in PA, suggesting possible benefits of living close to recreational facilities to sustain an active lifestyle [6]. Notably, additional cross-sectional analyses of MESA participants showed that living in areas with greater PA resources and access to healthy foods were also associated with lower insulin resistance [7] and lower incidence of type 2 diabetes mellitus [8].

The perceived environment is another relevant factor that may be associated with health outcomes. In Chicago, perceiving the neighborhood as safe was positively associated with walking levels, while perceived lower violence was associated with higher levels of leisure walking. However, in the same study, no significant associations were identified between perceived safety or police-recorded measures of crime and leisure PA [9]. To date, no analyses of the MESA cohort have investigated the associations of perceived environment and longitudinal patterns of PA.

Given this, our study aimed to assess if the perceived environment is significantly associated with longitudinal patterns of PA. We hypothesized that better perceptions of the environment would be associated with the adoption or maintenance of PA over time, while worse perceptions of the environment would be associated with being insufficiently active or not maintaining sufficient PA behavior over time.

## 2. Methods

### 2.1. Study Design

The current study is an analysis of MESA data. The MESA is a multi-site, prospective cohort in the United States that included men and women, free of known clinical cardiovascular disease, aged 45–84 years old at baseline, residing in one of the site locations: New York, New York; Baltimore, Maryland; Chicago, Illinois; Los Angeles, California; St. Paul, Minnesota; and Forsyth County, North Carolina. The Institutional Review Boards from all participating institutions approved the study, and written informed consent was obtained. Detailed information about the MESA study can be found elsewhere [10].

### 2.2. Participants

Participants were enrolled in the study between July 2000 and August 2002 (baseline visit) and returned for follow-up visits approximately every two years. MESA is composed of diverse ethnic backgrounds: White (38%), African American (28%), Hispanic (23%), and Asian (mostly Chinese American) (11%) individuals [10]. MESA participants who had available data for leisure-time physical activity at Exam 6 and at the previous exams with the exception of Exam 4 when PA was not assessed (n = 3097).

### 2.3. Exposure

The primary exposure variables were measured by questionnaire on the perceived aspects of the neighborhood environment at baseline. One question concerning safety asked, “How safe from crime do you consider your neighborhood to be?”, and participants rated their perception on a scale from 1–5, being 1 “very safe”, 3 “safe” and 5 “not at all safe”. A second question asked, “Think about your neighborhood as a whole, then please check one box for each of the following to show how much of a problem each one is in your neighborhood”. The items are excessive noise, heavy traffic or speeding cars, lack of access to adequate food shopping, lack of parks or playgrounds, trash or litter, no sidewalk or poorly maintained sidewalks, and violence. For each item, the response options were: very serious problem (1), somewhat serious problem (2), minor problem (3), not really a problem (4).

### 2.4. Outcome

At baseline and the subsequent study visits (except visit 4), physical activity (PA) was assessed using the MESA Typical Week Physical Activity Survey, adapted from the Cross-Cultural Activity Participation study [10,11]. We defined intentional exercise as the sum of walking for exercise, playing sports, dancing, and conditioning exercise, expressed in metabolic equivalents of a task (MET) per min/week. We classified participants according to the Physical Activity Guidelines for Adult Americans [12] as meeting or not meeting the recommendations (defined as an engagement in at least 150 min of moderate-to-vigorous-intensity PA (MVPA) per week). Because the intentional exercise was expressed as MET-minutes per week, the equivalent of 150 min/week of MVPA equals the range of 500–1000 MET-minutes/week [13]. Therefore, we considered the cut point of 500 MET-minutes/week to categorize individuals meeting (≥500 MET-minutes/week) versus not meeting the guidelines (<500 MET-minutes/week).

### 2.5. Covariates

At the clinic visits, standardized questionnaires were used to collect information on participants’ sociodemographic characteristics. These included age, sex, race/ethnicity, household assets, educational level, marital status (married/living as married versus other, which included widowed, divorced, separated, never married, and individuals who preferred not to answer), occupation/employment, and city of residency. An additional covariate, neighborhood-level socioeconomic status (SES), was available. This variable used 2000 U.S Census estimates linked to residential data of MESA participants [14]. A summary SES was built by factor analysis of six indicators of neighborhood-level SES, including the median household income, household wealth (median value of housing units and percent of households with interest, dividend, or net rental income), education (the percentage of adults who completed high school and the percentage of adults who completed college education), percentage of employment among people aged 16 years or older in an executive, managerial or professional occupation.

During the clinic visits, participants completed a health history questionnaire, which included questions on current alcohol consumption and smoking habits (never, former, or current smoker). Chronic diseases were defined as follows: (1) diagnosis of diabetes mellitus type II according to the American Diabetes Association algorithm published in 2003 [15] and (2) diagnosis of hypertension by the JNC VI (1997) criteria as normal (<130/<85 mmHg), high-normal (130–139/85–89 mmHg), stage 1 hypertension (140–159/90–99 mmHg), or stage 2 or greater hypertension (≥160/≥100 mmHg) [16]. Other self-reported chronic diseases were emphysema, asthma, and arthritis. Additionally, physical symptoms that could interfere with physical activity were self-reported pain in the lower limbs (“Do you ever get leg pain in either leg or buttock while walking?”) and swelling of feet and ankles (“Have you ever had swelling of your feet and ankles?”).

Anthropometric measures were taken with height, and weight measured to the nearest 0.1 cm and 0.5 kg, respectively, and the body mass index (BMI) was calculated (kg/m^2^). Waist circumference was assessed at the umbilicus, and the hip circumference was assessed at the maximal circumference of the buttocks using a steel measuring tape (standard 4 oz. tension) to the nearest 0.1 cm. Blood pressure was assessed in the right arm after five minutes of the participant resting in a sitting position. An automated oscillometric method (model Dinamap, GE Medical Systems Information Technologies, Inc., Milwaukee, Wisconsin, USA) and appropriate cuff size were used. Three readings were taken, and the average between the last two readings was considered for analyses. Fasting blood samples (75 mL) were drawn and used to determine the levels of low-density lipoprotein (LDL) cholesterol, high-density lipoprotein (HDL) cholesterol, total cholesterol, and triglycerides. These were categorized according to the National Cholesterol Education Program (NCEP) report [17].

### 2.6. Statistical Analyses

We created descriptive statistics, including means and standard deviations (SDs) or medians and interquartile ranges (IQRs) for continuous variables and frequencies for categorical variables. Physical activity was presented for all exams in which it was assessed (except Exam 4), including the prevalence of participants meeting the recommended guidelines of at least 150 min/week of moderate to vigorous PA (MVPA) [12]. We conducted an additional analysis comparing the characteristics of individuals at baseline who were part of our analytical sample (with complete data available at baseline and Exam 6) with individuals who were excluded from the analysis (due to incomplete data, mortality, or excluded for another reason). For these analyses, we tested differences between groups using independent *t*-tests for continuous variables and the Chi-square test for categorical variables.

We created categories and classified participants according to their physical activity behavior into the following groups: adopters (those who did not meet guidelines at baseline but met guidelines at Exam 6), relapsers (individuals who met the guidelines at baseline but did not meet guidelines at Exam 6), maintainers (individuals who met guidelines both at baseline and Exam 6), and insufficiently active (individuals who did not meet the guidelines at either baseline or Exam 6). We considered the maintainers’ group as the reference category in our analyses.

We conducted modified Poisson multinomial regression models to estimate the risk ratio (RR) according to methods proposed by Zou G [18]. The absolute risk differences were calculated according to each exposure variable. The models were adjusted sequentially where Model 1 was adjusted for the contextual level variables (study site and contextual markers of socioeconomic status [SES]); Model 2 was further adjusted for individual-level sociodemographic variables (age, sex, race/ethnicity, educational level, marital status, occupation); and Model 3 was further adjusted for individual-level health variables (obesity assessed through waist circumference, LDL cholesterol, HDL cholesterol, and triglyceride levels, hypertension diagnosis, diabetes diagnosis, smoking status, alcohol consumption, emphysema, asthma, arthritis, pain in the lower limbs, and swelling of the feet and ankles). Figure 1 below provides a visual description of the multi-level adjustment.

## 3. Results

At baseline, 6814 individuals were enrolled and evaluated. The analytical sample for our study includes 3097 of these individuals who had data available for intentional exercise at Exam 6. The characteristics of individuals included in the analytical sample are detailed in Table 1. They were, on average, 57.9 years old, 52.9% female, and 39.9% White. Most had completed high school or less and were married and employed.

Table 2 presents the perceptions of the neighborhood environment. At baseline, a small proportion of individuals perceived elements in the environment as a “very serious problem”, with frequencies being below 10% for all other exposures—excessive noise, traffic and speeding cars, no access to adequate food shopping, lacking parks and playgrounds, trash and litter, lack of or poor sidewalks, and violence. Excessive noise was mostly perceived as “not being really a problem” or a “minor problem”. Heavy traffic or speeding cars were mostly perceived as “not really a problem” or a “minor problem”. The lack of access to adequate food shopping and lacking parks and playgrounds were perceived as “not really a problem”. Most individuals perceived their neighborhoods as “safe”, followed by “more than safe”.

Intentional exercise across multiple time points is shown in Appendix C., along with the median level of intentional PA reported as METs-min/week and the prevalence of individuals meeting PA guidelines per Exam (≥500 METS-min/week). At baseline and Exam 6, the median values for intentional exercise were 900 (IQR: 210–2130) and 945 (IQR: 157.5–2280) METs-min/week, respectively. Self-reported intentional exercise was higher at Exam 5, compared to other time points (1860; IQR: 802.5–3780) METs-min/week. At this time point, most individuals reported enough PA to meet physical activity guidelines (82.8%). Categories created to discriminate patterns of PA were distributed as follows: 46.5% were classified as maintainers, 17.0% were adopters, 17.7% were relapsers, and 18.6% were insufficiently active.

We performed additional analyses to assess potential differences in baseline characteristics of individuals who had missing data versus those who answered the PA questionnaire at Exam 6. Appendix A summarizes the overall sample characteristics and details the two groups (included and excluded in the analyses). The two groups were not meaningfully different in terms of distribution of sex, BMI, diastolic blood pressure, the prevalence of diabetes, total cholesterol, and HDL cholesterol. Included individuals were younger, and there was a higher prevalence of participants of high SES compared to excluded participants. Regarding the individual-level characteristics, more White and Chinese individuals and fewer Black and Latino individuals composed the included sample. Included participants were also more educated and more likely to be married.

Participants included in the analysis had lower waist circumference, a lower mean systolic blood pressure, a lower prevalence of hypertension diagnosis, and lower mean triglycerides. Included participants also had a lower prevalence of emphysema, arthritis, pain in the legs or buttocks, and swelling of the feet and ankles. On the other hand, included participants had a higher prevalence of asthma. Regarding health behaviors, there was a lower prevalence of current smokers and higher current consumption of alcoholic drinks among the included participants.

No differences were identified for the perception of lacking adequate access to food shopping, the perception of lacking parks and playgrounds, and the perception of poor sidewalks between included and excluded participants. There were statistically significant differences in the distribution of frequencies for the perception of excessive noise, perception of heavy traffic and speeding cars, perception of trash or litter, perception of violence, and perception of safety between included and excluded participants. A higher proportion of included participants perceived excessive noise, traffic and speeding cars, trash or litter, and violence as a “minor problem” compared to excluded participants. Additionally, a smaller proportion of included participants perceived excessive noise, traffic and speeding cars, trash or litter, and violence as “not really a problem” compared to excluded participants. Lastly, a smaller proportion of included participants perceived the neighborhood as safe.

We performed a sensitivity analysis comparing the perception of the environment among individuals (Appendix B) by the multinomial regression models shown. No statistically significant associations between the perception of excessive noise, perception of heavy traffic and speeding cars, perception of the lack of access to adequate food shopping, perception of the presence of trash or litter, and perceived violence with longitudinal patterns of PA.

Individuals who reported that lack of parks and playgrounds was “not a problem” in their neighborhood had a 2.3-times higher risk of decreasing their physical activity (i.e., “relapser” category) compared to maintainers. There were no significant associations between perceptions of the lack of parks and playgrounds in the neighborhood and being categorized as adopters or insufficiently active (Table 3).

We observed adopters were less likely than maintainers to report perceiving lack of or poor sidewalks as “somewhat serious problem”. That is, and after full adjustment, to perceive poor sidewalks as “somewhat a serious problem” was associated with a 64% lower risk of becoming an adopter than maintainer. There were no significant associations seen for “relapsers” and “insufficiently active” categories (Table 4).

When compared to those who perceive the neighborhood as “very safe”, perception of the neighborhood as “safe” to “not at all safe” (rating 3, 4, and 5, respectively, in the perceived safety scale) was significantly associated with being classified in the adopter category. Additionally, and when compared to the same reference group, individuals who perceived the neighborhood as “safe” (rating 3) or as category 4 in the safety rating had a 1.5 and 1.8, respectively, higher risk of being categorized as insufficiently active. No significant associations were observed for individuals categorized as “relapsers” (Table 5).

## 4. Discussion

Our analyses suggest that perceived lack of parks and playgrounds, perception of no sidewalks or poorly maintained sidewalks, and perceived safety were associated with patterns of PA. Specifically, perceiving the lack of sidewalks or poorly maintained ones as “somewhat a serious problem” was associated with a lower risk of “adopting” PA over time. Additionally, we demonstrated that a perceived lack of safety was associated with being consistently insufficiently active over time. Of note, we also identified significant associations contrary to our hypothesis. That is, the perceived lack of parks/playgrounds as “not problematic” was associated with relapsing PA, and the perceived lack of safety was also associated with being an adopter of PA. We demonstrated no significant associations between perceived excessive noise, heavy traffic and speeding cars, lack of access to adequate food shopping, presence of trash or litter, and perceived violence with patterns of PA. Taken together, these results suggest that problematic perception of poor sidewalks is associated with lower rates of PA adoption and that perceived lack of safety is associated with sustaining insufficient PA levels.

We identified that the perception of the lack of parks and playgrounds as “not being problematic” increased the risk of an individual being a relapser (compared to those who maintained PA over time), which contradicted our initial hypothesis that perceiving the lack of parks and playgrounds as problematic would be associated with being a relapser or insufficiently active. A previous study from MESA that objectively assessed the density of recreational facilities identified a greater increase in density was associated with a lesser decline in physical activity over time [6] after adjustment for the individual-level perception of the environment. Still, Ranchod at al. [6] did not assess the perceived environment alone and its association with levels of PA. Moreover, a cross-sectional analysis from the International Physical Activity and Environment Network (IPEN) study demonstrated that the number of parks in the neighborhood was associated with higher levels of PA [19], while a cross-sectional analysis of participants from Australia showed that non-retired individuals reporting living near a park were more likely to participate in recreational walking [20] but not other types of recreational MVPA. Given the difference between our results and these and others, additional longitudinal analyses are warranted to understand better the role of the perception of parks and engagement in intentional PA.

Our study showed that adopters were less likely than maintainers to report perceiving the lack of sidewalks or poorly maintained ones as “somewhat a serious problem”. Our findings are supported by the previous literature regarding the perception of sidewalks in other countries [21,22]. In the U.S, the association was not statistically significant [21]. In summary, the presence of sidewalks, perceived adequate esthetics, and evenness of sidewalks evident in the literature corroborate our findings [22,23]. It is noteworthy that previous literature conducted on this theme within the U.S is cross-sectional, and our longitudinal design demonstrates that this association is consistent independent of contextual SES and individual-level characteristics.

Perceptions of an unsafe neighborhood were associated with being classified as an “adopter” of PA and “insufficiently active”. The former contradicts our hypothesis, while the association between perceived lack of safety and being insufficiently active aligns with our hypothesis. Within MESA, a previous cross-sectional analysis identified that perceiving a safe neighborhood was positively associated with transport walking but not with leisure walking or intentional PA engagement [9]. The previous literature has also highlighted inconsistent findings on the association between perceived safety and PA engagement [24,25,26,27].

Our study has strengths and limitations. Strengths include a large multi-ethnic sample size and a longitudinal design. Moreover, our study has a long period between assessments (approximately 16 to 18 years), which can also be pointed out as a limitation, as sparse data can increase the probability of residual confounding in our analyses. As for limitations, our inclusion criteria may have introduced selection bias, and both the exposure and the physical activity levels were self-reported. Differences between self-reported and objectively measured PA have been demonstrated in the literature [28,29]. Additionally, we considered intentional PA, and we did not include other domains of PA (e.g., transportation, activities of daily living). Additionally, we did not adjust our analyses for individuals who may have moved within the period analyzed.

The main implication for practice of our findings includes the awareness that perceived lack of or poor sidewalks was associated with prevention of PA adoption, and that perceived lack of safety was associated with insufficient PA behavior independently of contextual and individual-level factors. Our study can inform policymakers and professionals involved in developing residential areas by emphasizing adequate sidewalks and aspects that can increase the perception of safety.

We recommend objectively measured physical activity to decrease potential recall bias. Future analyses should also consider that participants may have moved during the period analyzed. Therefore, moving patterns should be considered. Additionally, physical activity from a one-time point to the following be considered in addition to the overall time analyzed (baseline to Exam 6). The development of chronic diseases, physical symptoms, and limitations during the period analyzed could influence one’s ability to engage in intentional physical activity and should also be considered. We also recommend that future studies consider other domains of PA. Finally, objective measures of the environment and perceptions of it should be used in combination [29] to understand better how the environment relates to patterns of PA.

## 5. Conclusions

Aspects of the perceived environment are associated with being physically active in adults over approximately 18 years. Problematic perception of the lack of, or poorly maintained, sidewalks was associated with adopting PA guidelines, and perceived lack of safety was associated with insufficiently active behavior. Such findings should be considered in public health initiatives to promote physical activity.

## Figures and Tables

**Figure 1 ijerph-19-10925-f001:**
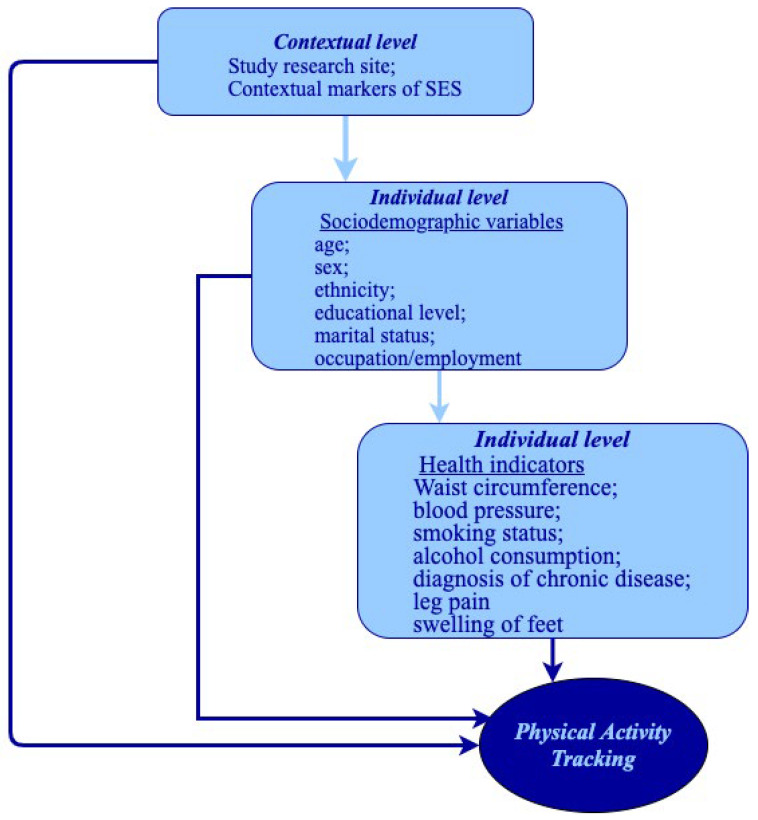
Conceptual multi-level framework of exposure effects on physical activity patterns.

**Table 1 ijerph-19-10925-t001:** Characteristics of participants at baseline (n = 3097) expressed as mean (95% CI) or frequencies (95% CI) (continued).

Characteristics	n	
**Study Site (%)** Winston-Salem, NC New York, NY Baltimore, MD Twin Cities, MN Chicago, IL Los Angeles, CA	3097	11.62 (10.54, 12.80) 17.99 (16.77, 19.38) 13.63 (12.46, 14.88) 17.95 (16.64, 19.35) 20.96 (19.56, 22.43) 17.86 (16.55, 19.55)
**Age (years)**	3097	57.96 (57.65, 58.26)
**Sex (%)** Male Female	3097	47.11 (51.13, 54.64) 52.89 (45.36, 48.87)
**Contextual marker of SES (%)** Low SES Medium SES High SES	3064	35.61 (33.93, 37.32) 29.01 (27.43, 30.65) 35.38 (33.70, 37.09)
**Race/ethnicity (%)** White Asian (mostly Chinese American) African American/Black Hispanic/Latino	3097	39.97 (38.26, 41.71) 13.27 (12.12, 14.51) 25.22 (23.72, 26.78) 21.54 (20.12, 23.02)
**Education (%)** High school or less Incomplete or technical school College degree Graduate degree	3091	28.28 (26.71, 29.89) 23.75 (22.28, 25.28) 24.91 (23.42, 26.47) 23.07 (21.61, 24.59)
**Marital status (%)** Married/living as married Other	3091	65.22 (63.52, 66.88) 34.78 (33.12, 36.48)
**Occupation (%)** Employed full-time/homemaker Employed part-time Unemployed/on leave Retired	3091	61.47 (59.74, 63.17) 10.32 (9.30, 11.44) 3.53 (2.93, 4.24) 24.68 (23.20, 26.24)
**BMI (kg/m^2^)**	3097	28.20 (28.01, 28.39)
**Waist circumference (cm)**	3097	96.83 (96.33, 97.32)
**Blood pressure(mmHg)** Systolic Diastolic	3097	121.75 (121.06, 122.45) 71.84 (71.49, 72.20)
**Hypertension diagnosis (%)**	3097	34.90 (0.33, 0.37)
**Diabetes type II (%)**	3097	6.23 (5.43, 7.14)
**Total cholesterol (mg/dL)** High, ≥240 mg/dL	3086	194.82 (193.58, 196.06) 9.49 (8.51, 10.58)
**HDL Cholesterol (mg/dL)** Low, <40 mg/dL (%)	3086	51.06 (50.54, 51.58) 21.08 (20.26, 23.17)
**LDL Cholesterol (mg/dL)** Borderline high, 130–159 (%) High, 160–189 (%) Very high, ≥190 (%)	3056	118.18 (31.14) 24.21 (22.73, 25.77) 7.30 (6.43, 8.28) 1.83 (1.41, 2.37)
**Triglycerides (mg/dL)** Borderline high, 150–199 (%) High, 200–499 (%) Very high, ≥500 (5%)	3086	128.79 (125.97, 131.61) 15.23 (14.00, 16.54) 13.29 (12.13, 14.53) 0.39 (0.22, 0.68)
**Smoking (%)** Never Former Current	3091	52.86 (51.10, 54.62) 35.78 (34.11, 37.49) 11.36 (10.28, 12.52)
**Alcohol consumption (%)** Never Former Current	3080	18.57 (17.14, 19.88) 19.97 (18.59, 21.42) 61.56 (59.83, 63.26)
**Emphysema (%)**	3096	0.84 (0.57, 1.23)
**Asthma (%)**	3095	10.15 (9.13, 11.26)
**Arthritis (%)**	3096	28.20 (26.64, 29.81)
**Leg or buttock pain (%)**	3096	22.71 (21.26. 24.22)
**Swelling of feet or ankle (%)**	3094	27.44 (25.90, 29.04)

BMI: body mass index; HDL: high-density lipoprotein; LDL: low-density lipoprotein; SES: socioeconomic status.

**Table 2 ijerph-19-10925-t002:** Perceptions of the neighborhood environment at baseline expressed in frequencies (95% CI).

		**n**	**Very Serious Problem**	**Somewhat Serious Problem**	**Minor Problem**	**Not Really a Problem**
**Excessive noise**		3092	3.62 (3.02, 4.34)	12.48 (11.36, 3.70)	35.87 (34.19, 37.57)	48.03 (46.27, 49.79)
**Traffic/speeding cars**		3092	6.18 (5.38, 7.08)	16.95 (15.66, 18.31)	34.54 (32.88, 36.24)	42.34 (40.60, 44.09)
**No access to food shopping**		3093	1.26 (0.92, 1.72)	3.72 (3.11, 4.45)	14.03 (12.85, 15.30)	80.99 (79.57, 82.33)
**Lacking parks and playgrounds**		3084	2.4 (1.91, 3.00)	5.03 (4.31, 5.86)	16.37 (15.11, 17.72)	76.20 (74.66, 77.67)
**Trash and litter**		3087	3.14 (2.58, 3.82)	7.13 (6.27, 8.09)	27.79 (26.94, 29.40)	61.94 (60.21, 63.64)
**Poor sidewalks**		3087	1.98 (1.54, 2.53)	4.18 (3.53, 4.94)	17.23 (15.94, 18.61)	76.61 (75.08, 78.07)
**Violence**		3087	1.81 (1.40, 2.35)	8.13 (7.22, 9.15)	24.46 (22.97, 26.01)	65.50 (63.90, 67.25)
	**n**	**(5)** **Not at all safe**	**(4)**	**(3)** **Safe**	**(2)**	**(1)** **Very safe**
Safety	3082	3.21 (2.64, 3.90)	12.46 (11.34, 13.67)	44.78 (43.03, 46.54)	19.73 (18.36, 21.17)	19.92 (18.45, 21.27)

**Table 3 ijerph-19-10925-t003:** Multinomial regression models assessing the association between the perception of lack of parks and playgrounds in the neighborhood and patterns of PA (continued).

		Model 1 (n = 3050)	Model 2 (n = 3048)	Model 3 (n = 3023)
		RR 95% CI	RR 95% CI	RR 95% CI
Maintainers (Ref)		-	-	-
Adopters	Serious problem (ref)	**-**	**-**	**-**
	Somewhat serious	1.97 (0.81, 4.79)	1.88 (0.77, 4.62)	1.99 (0.80, 4.95)
	Minor problem	1.62 (0.72, 3.66)	1.50 (0.66, 3.41)	1.57 (0.68, 3.60)
	Not a problem	1.78 (0.81, 3.89)	1.72 (0.78, 3.80)	1.82 (0.82, 4.06)
Relapsers	Serious problem (ref)	-	-	-
	Somewhat serious	1.38 (0.57, 3.33)	1.72 (0.69, 4.32)	1.91 (0.75, 4.89)
	Minor problem	1.58 (0.72, 3.43)	1.93 (0.85, 4.37)	2.07 (0.90, 4.78)
	Not a problem	1.74 (0.83, 3.69)	2.01 (0.91, 4.42)	**2.29** **(1.02, 5.14)**
Insufficiently Active	Serious problem (ref)	-	-	-
	Somewhat serious	0.96 (0.48, 1.92)	1.04 (0.51, 2.11)	1.26 (0.602, 2.63)
	Minor problem	0.75 (0.41, 1.38)	0.81 (0.44, 1.52)	0.97 (0.51, 1.86)
	Not a problem	0.80 (0.45, 1.42)	0.86 (0.48, 1.54)	1.08 (0.59, 2.00)

Model 1: adjusted for study site and contextual markers of SES; Model 2: model 1 + age, sex, race/ethnicity, educational level, marital status, and occupation; Model 3: model 2 + waist circumference, LDL cholesterol, HDL cholesterol, triglycerides, hypertension, diabetes, smoking, alcohol, emphysema, asthma, arthritis, pain in the lower limbs, and swelling of feet and ankles. CI: confidence interval; RR: risk ratio; PA: physical activity. **Significant association shown in bold**.

**Table 4 ijerph-19-10925-t004:** Multinomial regression models assessing the association between the perception of poor sidewalks in the neighborhood and patterns of PA.

		Model 1 (n = 3086)	Model 2 (n = 3053)	Model 3 (n = 3051)
		RR 95% CI	RR 95% CI	RR 95% CI
Maintainers (Ref)		-	-	-
Adopters	Serious problem (ref)	**-**	**-**	**-**
	Somewhat serious	**0.37** **(0.15, 0.91)**	**0.38** **(0.15, 0.96)**	**0.36** **(0.14, 0.93)**
	Minor problem	0.74 (0.35, 1.58)	**0.77** **(0.15, 0.96)**	0.76 (0.35, 1.66)
	Not a problem	0.68 (0.33, 1.40)	0.72 (0.34, 1.50)	0.72 (0.34, 1.52)
Relapsers	Serious problem (ref)	-	-	-
	Somewhat serious	0.48 (0.20, 1.18)	0.52 (0.21, 1.28)	0.49 (0.19, 1.22)
	Minor problem	0.79 (0.37, 1.70)	0.88 (0.41, 1.92)	0.88 (0.40, 1.94)
	Not a problem	0.68 (0.33, 1.43)	0.73 (0.34, 1.54)	0.74 (0.34, 1.58)
Insufficiently Active	Serious problem (ref)	-	-	-
	Somewhat serious	0.47 (0.21, 1.08)	0.57 (0.25, 1.34)	0.56 (0.23, 1.33)
	Minor problem	0.64 (0.31, 1.08)	0.79 (0.38, 1.66)	0.83 (0.39, 1.77)
	Not a problem	0.60 (0.30, 1.20)	0.70 (0.35, 1.43)	0.76 (0.37, 1.58)

Model 1: adjusted for study site and contextual markers of SES; Model 2: model 1 + age, sex, race/ethnicity, educational level, marital status, and occupation; Model 3: model 2 + waist circumference, LDL cholesterol, HDL cholesterol, triglycerides, hypertension, diabetes, smoking, alcohol, emphysema, asthma, arthritis, pain in the lower limbs, and swelling of feet and ankles. CI: confidence interval; RR: risk ratio; PA: physical activity. **Significant association shown in bold**.

**Table 5 ijerph-19-10925-t005:** Multinomial regression models assessing the association of perceived safety in the neighborhood and patterns of PA.

	Model 1 (n = 3048)	Model 2 (n = 3046)	Model 3 (n = 3021)
	RR 95% CI	RR 95% CI	RR 95% CI
Maintainers (Ref)		-	-
Very safe (1) (ref)	**-**	**-**	**-**
(2)	1.16 (0.83, 1.62)	1.23 (0.87, 1.73)	1.27 (0.90, 1.80)
Safe (3)	**1.51** **(1.11, 2.04)**	**1.56** **(1.15, 2.13)**	**1.62** **(1.18, 2.22)**
(4)	**1.50** **(1.00, 2.23)**	**1.61** **(1.07, 2.44)**	**1.66** **(1.09, 2.52)**
Not at all safe (5)	**2.10** **(1.10, 4.00)**	**2.20** **(1.14, 4.26)**	**2.14** **(1.10, 4.17)**
Very safe (1) (ref)	**-**	**-**	**-**
(2)	0.88 (0.64, 1.21)	0.91 (0.65, 1.26)	0.92 (0.66, 1.29)
Safe (3)	1.14 (0.86, 1.52)	1.08 (0.81, 1.44)	1.07 (0.79, 1.43)
(4)	1.05 (0.71, 1.56)	1.00 (0.67, 1.49)	1.03 (0.68, 1.55)
Not at all safe (5)	1.80 (0.98, 3.29)	1.45 (0.78, 2.68)	1.23 (0.65, 2.31)
Very safe (1) (ref)	-	-	-
(2)	0.84 (0.59, 1.19)	0.93 (0.65, 1.33)	0.97 (0.67, 1.39)
Safe (3)	**1.48** **(1.10, 2.00)**	**1.45** **(1.07, 1.96)**	**1.48** **(1.09, 2.02)**
(4)	**1.78** **(1.22, 2.59)**	**1.78** **(1.20, 2.63)**	**1.81** **(1.21, 2.70)**
Not at all safe (5)	**2.03** **(1.10, 3.74)**	1.64 (0.87, 3.06)	1.43 (0.75, 2.71)

Model 1: adjusted for study site and contextual markers of SES; Model 2: model 1 + age, sex, race/ethnicity, educational level, marital status, and occupation; Model 3: model 2 + waist circumference, LDL cholesterol, HDL cholesterol, triglycerides, hypertension, diabetes, smoking, alcohol, emphysema, asthma, arthritis, pain in the lower limbs and swelling of feet and ankles. CI: confidence interval; RR: risk ratio; PA: physical activity. **Significant association shown in bold**.

## Data Availability

The datasets supporting the conclusions of this article can be accessed by reasonable request to MESA Publication and Presentations Committee (https://www.mesa-nhlbi.org, accessed in 1 June 2021).

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
