# Peer review of "Longitudinal Associations of Physical Activity Patterns and the Environment: An 18-Year Follow-Up to the MESA Study"

_ijerph, 2022, doi:10.3390/ijerph191710925_

Round 1

Reviewer 1 Report

This is an interesting study with some potentially useful findings. It will likely merit publication after minor revision to correct errors in presentation, clarify methods and findings, and other issues identified below.

Abstract (lines 20-35): The abstract is vague and should be completely rewritten to correct grammatical errors, add information on methods, and more clearly state the major findings as listed in results and especially in conclusions. Some of the issues in the abstract are:

Lines 23-25: The statement beginning “To assess” is an incomplete sentence. Suggest add an introductory phrase to correct the problem such as: "The present study was undertaken to assess.... and evaluate...."

 Lines 25-26: The initial statement following Methods is also an incomplete sentence and factually misleading. While the total number of participants in the MESA study was 6,814, only a 3,097- person subset was used in this study. This subset was selected based on the availability of data on intentional PA for each participant. The sentence should be restated to reflect this information.

 Lines 28-29 should be revised to very briefly explain what your categories mean (e.g., adopters means those who started PA after the MESA study was initiated), etc.

 Lines 30-35: Results and conclusions – I suggest you use the next to last paragraph in the Results section here as, with the last sentence perhaps added to your current conclusions statement in the abstract.

 Other comments

 General comment regarding the Introduction and Discussion: The authors have provided less review of potentially relevant literature than I expected. A cursory google search yielded several papers that are not cited here, including one longitudinal study, and I expect that there are many more. I strongly urge the authors to do a more thorough review of the literature, especially if they plan to continue to include a statement that indicates that their study is the first of its kind.

 Potentially relevant papers (these are included as potential papers to consider, not recommendations). The authors are encouraged to do a more thorough literature search.

Kwarteng et al. 2014. Journal of Public Health 36(3): 358-367.  

Christian et al. 2016. Environ. Health Persp. Doi.org/10.1289/EHP823. This is a longitudinal study.

 Berger et al. 2019. BMC Public Health doi.org/10.1186/s12889-019-8003-7

 Mooney et al. 2017. J. Urban Health 94. Doi.org/10.1007/s11525-016-0125-y

 Liu et al. 2021. Front. Public Health  https://doi.org/10.1289/EHP823.

 Line 41: Insert “activity” after physical

 Line 47 and thereafter: I believe citation numbers should be included in brackets, not parentheses.

 Lines 71-73: You may wish to include a shorter version of this statement near the end the Results section in your abstract, if you can demonstrate that it is the first such study. For example, “As the first longitudinal study of association of perceived environmental characteristics and physical activity, this study showed….”  This is truly a minor suggestion, but if yours is the first such study you might want to get that up front. If there are any questions about it being the first, you may wish to use a more equivocal statement such as “one of the first or one of the few”.

 Lines 91-92: This statement about diversity of MESA participants seems out of place, considering that you have more text on the demographic data associated with your cohort later in the paper. I suggest delete it here.

 Lines 93-94: This sentence is confusing. It says that you chose participants based on having PA data at Exam 6, but I believe you mean people for whom PA data were available for Exam 6 and all previous time periods, with the exception of period 4 when PA data were not collected.  Some clarification here would make the claim of a longitudinal study clear early in the paper. Inclusion of Appendix 3 as a table within the text, perhaps near here (see later comment), would also be useful in this regard.

 Lines 110-115: Were these defining elements of PA the same as used in the MESA study and in other publications using MESA data or do they differ in any meaningful way?

 Line 115: Suggest insert (MVPA) following PA in this line, since you use the abbreviation later in the paper.

 Line 179: How were the models adjusted?

 Line 191: What year was Exam 6, 2018?

 Lines 192-193: I do not think that 53% equates to “mostly female.” I suggest that you summarize all the demographic information here in one paragraph and then perhaps provide a similar paragraph for the health data. For example, “Study participants were not evenly distributed among geographic sites, but there were no pronounced differences in the proportions related to site and average age of members was 58 yrs. There were slightly more females than males, Whites composed 40% of the population, while Blacks, Asians (mostly Chinese), and Hispanics together constituted 60% of the sample, and participants were nearly evenly apportioned among low, medium, and high socio-economic status (Table 1).” This is just an example of how you might draft a paragraph that summarizes the demographic data. It is not suggested as a preferred way to do so.  More sentences should be included as needed to summarize other demographic data as you feel appropriate as well as health data.

 Line 197: Suggest replace semi-colon with colon before the list of exposures.

 Line 199: Revise “lack of poor sidewalks” to “lack of or poor sidewalks” as stated elsewhere in the paper.

Line 207: Supplementary Table 1 is referenced here. However, there is no such table in the Supplemental Materials. Instead, there are three appendices and I believe this reference is to Appendix 3. Since Appendix 3 is a small table and has some data that may be important for readers to review, I suggest include it in the paper text as noted above.

 Lines 254-256: Appendix 3 is mentioned here. This sentence seems out of place in this paragraph that talks primarily about parks and playgrounds. I suggest relocate it to the Methods section or early in the Results.

 Line 272: Suggest delete “The findings of” and begin sentence with “Our analyses suggest…”

 Line 294: The IPEN study is not previously mentioned. Please define the abbreviation and provide a citation.

 Line 329:  Duplication of “poor sidewalks” in the sentence. Should be “perceived lack of or poor sidewalks”.

 Lines 328-333: Consider for restating or abbreviating in the Abstract.

 Line 342: Citation 39 appears to be incorrect. I assume it should be 29.

Author Response

Austin, Aug 26th, 2021.

Dear Editors

We are grateful for the opportunity to receive valuable suggestions from the reviewers. We tried to meet each point of improvement raised, leaving the changes made to the text highlighted with track changes. We provided an answer to each reviewer's commentaries below. We are open to new suggestions and comments that might improve our paper.

Response to Reviewer 1 Comments

Point 1: This is an interesting study with some potentially useful findings. It will likely merit publication after minor revision to correct errors in presentation, clarify methods and findings, and other issues identified below.

Abstract (lines 20-35): The abstract is vague and should be completely rewritten to correct grammatical errors, add information on methods, and more clearly state the major findings as listed in results and especially in conclusions. Some of the issues in the abstract are:

Lines 23-25: The statement beginning “To assess” is an incomplete sentence. Suggest add an introductory phrase to correct the problem such as: "The present study was undertaken to assess.... and evaluate...."

 Lines 25-26: The initial statement following Methods is also an incomplete sentence and factually misleading. While the total number of participants in the MESA study was 6,814, only a 3,097- person subset was used in this study. This subset was selected based on the availability of data on intentional PA for each participant. The sentence should be restated to reflect this information.

 Lines 28-29 should be revised to very briefly explain what your categories mean (e.g., adopters means those who started PA after the MESA study was initiated), etc.

 Lines 30-35: Results and conclusions – I suggest you use the next to last paragraph in the Results section here as, with the last sentence perhaps added to your current conclusions statement in the abstract.

Response: Your suggestion was accepted, we changed the abstract to aim, which is better for readers to understand this section.

Point 2: Other comments

 General comment regarding the Introduction and Discussion: The authors have provided less review of potentially relevant literature than I expected. A cursory google search yielded several papers that are not cited here, including one longitudinal study, and I expect that there are many more. I strongly urge the authors to do a more thorough review of the literature, especially if they plan to continue to include a statement that indicates that their study is the first of its kind.

 Potentially relevant papers (these are included as potential papers to consider, not recommendations). The authors are encouraged to do a more thorough literature search.

Kwarteng et al. 2014. Journal of Public Health 36(3): 358-367. 

Christian et al. 2016. Environ. Health Persp. Doi.org/10.1289/EHP823. This is a longitudinal study.

 Berger et al. 2019. BMC Public Health doi.org/10.1186/s12889-019-8003-7

 Mooney et al. 2017. J. Urban Health 94. Doi.org/10.1007/s11525-016-0125-y

 Liu et al. 2021. Front. Public Health  https://doi.org/10.1289/EHP823.

Response: Your suggestion was accepted; we improved the introduction and discussion. The reviewer can see all the changes in the new version of the manuscript (with track changes). 

Point 2: Line 41: Insert “activity” after physical

 Line 47 and thereafter: I believe citation numbers should be included in brackets, not parentheses.

 Lines 71-73: You may wish to include a shorter version of this statement near the end the Results section in your abstract, if you can demonstrate that it is the first such study. For example, “As the first longitudinal study of association of perceived environmental characteristics and physical activity, this study showed….”  This is truly a minor suggestion, but if yours is the first such study you might want to get that up front. If there are any questions about it being the first, you may wish to use a more equivocal statement such as “one of the first or one of the few”.

 Lines 91-92: This statement about diversity of MESA participants seems out of place, considering that you have more text on the demographic data associated with your cohort later in the paper. I suggest delete it here.

Lines 93-94: This sentence is confusing. It says that you chose participants based on having PA data at Exam 6, but I believe you mean people for whom PA data were available for Exam 6 and all previous time periods, with the exception of period 4 when PA data were not collected.  Some clarification here would make the claim of a longitudinal study clear early in the paper. Inclusion of Appendix 3 as a table within the text, perhaps near here (see later comment), would also be useful in this regard.

 Lines 110-115: Were these defining elements of PA the same as used in the MESA study and in other publications using MESA data or do they differ in any meaningful way?

 Line 115: Suggest insert (MVPA) following PA in this line, since you use the abbreviation later in the paper.

 Line 179: How were the models adjusted?

 Line 191: What year was Exam 6, 2018?

 Lines 192-193: I do not think that 53% equates to “mostly female.” I suggest that you summarize all the demographic information here in one paragraph and then perhaps provide a similar paragraph for the health data. For example, “Study participants were not evenly distributed among geographic sites, but there were no pronounced differences in the proportions related to site and average age of members was 58 yrs. There were slightly more females than males, Whites composed 40% of the population, while Blacks, Asians (mostly Chinese), and Hispanics together constituted 60% of the sample, and participants were nearly evenly apportioned among low, medium, and high socio-economic status (Table 1).” This is just an example of how you might draft a paragraph that summarizes the demographic data. It is not suggested as a preferred way to do so.  More sentences should be included as needed to summarize other demographic data as you feel appropriate as well as health data.

 Line 197: Suggest replace semi-colon with colon before the list of exposures.

 Line 199: Revise “lack of poor sidewalks” to “lack of or poor sidewalks” as stated elsewhere in the paper.

Line 207: Supplementary Table 1 is referenced here. However, there is no such table in the Supplemental Materials. Instead, there are three appendices and I believe this reference is to Appendix 3. Since Appendix 3 is a small table and has some data that may be important for readers to review, I suggest include it in the paper text as noted above.

 Lines 254-256: Appendix 3 is mentioned here. This sentence seems out of place in this paragraph that talks primarily about parks and playgrounds. I suggest relocate it to the Methods section or early in the Results.

 Line 272: Suggest delete “The findings of” and begin sentence with “Our analyses suggest…”

 Line 294: The IPEN study is not previously mentioned. Please define the abbreviation and provide a citation.

 Line 329:  Duplication of “poor sidewalks” in the sentence. Should be “perceived lack of or poor sidewalks”.

 Lines 328-333: Consider for restating or abbreviating in the Abstract.

 Line 342: Citation 39 appears to be incorrect. I assume it should be 29.

Response: Your suggestion was accepted; we did all the suggestions and corrected the typos that the reviewer mentioned here. Also, we improved the tables and results description for better understanding. The reviewer can see all the changes in the new version of the manuscript (with track changes). 

Yours sincerely,

Maíra Tristão Parra, ScD, MS, MPH

Assistant Project Scientist, Hebert Wertheim School of Public Health and Longevity Sciences

Assistant Manager at the Exercise and Physical Activity Resource Center, HWSPH

UC San Diego

Phone: 858-822-1740

e-mail: mtristaoparra@health.ucsd.edu

Augusto Cesar F. De Moraes, Ph.D., M.Sc., BS

Assistant Professor

Michael & Susan Dell Center for Healthy Living​

The University of Texas Health Science Center at Houston

UTHealth | School of Public Health Austin Campus

Department of Epidemiology, Human Genetics and Environmental Science

1616 Guadalupe | Suite 6.300 | Austin, TX 78701

e-mail: Augusto.DeMoraes@utm.tmc.edu

Reviewer 2 Report

(1) The chapter arrangement of the article should be adjusted appropriately. If tables are reduced(table 1), several graphs should be added at the same time. In the first row of Table 1, add a description of the numerical header.

(2) The Conclusion(line 344) is too simple and should be elaborated.

Author Response

Austin, Aug 26th, 2021.

Dear Editors

We are grateful for the opportunity to receive valuable suggestions from the reviewers. We tried to meet each point of improvement raised, leaving the changes made to the text highlighted with track changes. We provided an answer to each reviewer's commentaries below. We are open to new suggestions and comments that might improve our paper.

Response to Reviewer 2 Comments

Point 1: The chapter arrangement of the article should be adjusted appropriately. If tables are reduced (table 1, several graphs should be added at the same time. In the first row of Table 1, add a description of the numerical header.

Response: Your suggestion was accepted, we improved the tables and results description for better understanding.

Point 2: The Conclusion(line 344) is too simple and should be elaborated.

Response: Your suggestion was accepted; we improved the conclusion. The reviewer can see all the changes in the new version of the manuscript (with track changes). 

Point 2: Line 41: Insert “activity” after physical

 Line 47 and thereafter: I believe citation numbers should be included in brackets, not parentheses.

 Lines 71-73: You may wish to include a shorter version of this statement near the end the Results section in your abstract, if you can demonstrate that it is the first such study. For example, “As the first longitudinal study of association of perceived environmental characteristics and physical activity, this study showed….”  This is truly a minor suggestion, but if yours is the first such study you might want to get that up front. If there are any questions about it being the first, you may wish to use a more equivocal statement such as “one of the first or one of the few”.

 Lines 91-92: This statement about diversity of MESA participants seems out of place, considering that you have more text on the demographic data associated with your cohort later in the paper. I suggest delete it here.

Lines 93-94: This sentence is confusing. It says that you chose participants based on having PA data at Exam 6, but I believe you mean people for whom PA data were available for Exam 6 and all previous time periods, with the exception of period 4 when PA data were not collected.  Some clarification here would make the claim of a longitudinal study clear early in the paper. Inclusion of Appendix 3 as a table within the text, perhaps near here (see later comment), would also be useful in this regard.

 Lines 110-115: Were these defining elements of PA the same as used in the MESA study and in other publications using MESA data or do they differ in any meaningful way?

 Line 115: Suggest insert (MVPA) following PA in this line, since you use the abbreviation later in the paper.

 Line 179: How were the models adjusted?

 Line 191: What year was Exam 6, 2018?

 Lines 192-193: I do not think that 53% equates to “mostly female.” I suggest that you summarize all the demographic information here in one paragraph and then perhaps provide a similar paragraph for the health data. For example, “Study participants were not evenly distributed among geographic sites, but there were no pronounced differences in the proportions related to site and average age of members was 58 yrs. There were slightly more females than males, Whites composed 40% of the population, while Blacks, Asians (mostly Chinese), and Hispanics together constituted 60% of the sample, and participants were nearly evenly apportioned among low, medium, and high socio-economic status (Table 1).” This is just an example of how you might draft a paragraph that summarizes the demographic data. It is not suggested as a preferred way to do so.  More sentences should be included as needed to summarize other demographic data as you feel appropriate as well as health data.

 Line 197: Suggest replace semi-colon with colon before the list of exposures.

 Line 199: Revise “lack of poor sidewalks” to “lack of or poor sidewalks” as stated elsewhere in the paper.

Line 207: Supplementary Table 1 is referenced here. However, there is no such table in the Supplemental Materials. Instead, there are three appendices and I believe this reference is to Appendix 3. Since Appendix 3 is a small table and has some data that may be important for readers to review, I suggest include it in the paper text as noted above.

 Lines 254-256: Appendix 3 is mentioned here. This sentence seems out of place in this paragraph that talks primarily about parks and playgrounds. I suggest relocate it to the Methods section or early in the Results.

 Line 272: Suggest delete “The findings of” and begin sentence with “Our analyses suggest…”

 Line 294: The IPEN study is not previously mentioned. Please define the abbreviation and provide a citation.

 Line 329:  Duplication of “poor sidewalks” in the sentence. Should be “perceived lack of or poor sidewalks”.

 Lines 328-333: Consider for restating or abbreviating in the Abstract.

 Line 342: Citation 39 appears to be incorrect. I assume it should be 29.

Response: Your suggestion was accepted; we did all the suggestions and corrected the typos that the reviewer mentioned here. Also, we improved the tables and results description for better understanding. The reviewer can see all the changes in the new version of the manuscript (with track changes). 

Yours sincerely,

Maíra Tristão Parra, ScD, MS, MPH

Assistant Project Scientist, Hebert Wertheim School of Public Health and Longevity Sciences

Assistant Manager at the Exercise and Physical Activity Resource Center, HWSPH

UC San Diego

Phone: 858-822-1740

e-mail: mtristaoparra@health.ucsd.edu

Augusto Cesar F. De Moraes, Ph.D., M.Sc., BS

Assistant Professor

Michael & Susan Dell Center for Healthy Living​

The University of Texas Health Science Center at Houston

UTHealth | School of Public Health Austin Campus

Department of Epidemiology, Human Genetics and Environmental Science

1616 Guadalupe | Suite 6.300 | Austin, TX 78701

e-mail: Augusto.DeMoraes@utm.tmc.edu
